# ESTIMATION OF NUMBER OF COMMUNITIES IN ASSORTATIVE SPARSE NETWORKS

## ABSTRACT

Most community detection algorithms assume the number of communities, $K$, to be known *a priori*. Among various approaches that have been proposed to estimate $K$, the non-parametric methods based on the spectral properties of the Bethe Hessian matrices have garnered much popularity for their simplicity, computational efficiency, and robust performance irrespective of the sparsity of the input data. Recently, one such method has been shown to estimate $K$ consistently if the input network is generated from the (semi-dense) stochastic block model, when the average of the expected degrees ($\tilde{d}$) of all the nodes in the network satisfies $\tilde{d} \gg \log(N)$ ($N$ being the number of nodes in the network). In this paper, we prove some finite sample results that hold for $\tilde{d} = o(\log(N))$, which in turn show that the estimation of $K$ based on the spectra of the Bethe Hessian matrices is consistent not only for the semi-dense regime, but also for the sub-logarithmic sparse regime when $1 \ll \tilde{d} \ll \log(N)$. Thus, our estimation procedure is a robust method for a wide range of problem settings, regardless of the sparsity of the network input.

## 1 INTRODUCTION

Statistical analysis of network data has now become an extensively studied field within statistics and machine learning (see (Goldenberg et al., 2010; Kolaczyk & Csárdi, 2014; Newman, 2018) for reviews). Network datasets show up in several disciplines. Examples include networks originating from biosciences such as gene regulation networks (Emmert-Streib et al. (2014)), protein-protein interaction networks (De Las Rivas & Fontanillo (2010)), structural (Rubinov & Sporns (2010)) and functional networks (Friston (2011)) of brain and epidemiological networks (Reis et al. (2007)); networks originating from social media such as Facebook, Twitter and LinkedIn (Faloutsos et al. (2010)); citation and collaboration networks (Lehmann et al. (2003)); information and technological networks such as internet-based networks (Adamic & Glance (2005)), power networks (Pagani & Aiello (2013)) and cell-tower networks (Isaacman et al. (2011)). There are several active areas of research in developing statistical methodologies for network data analysis and also deriving the theoretical properties of the statistical methods. *In this paper, we focus on networks with community structure and finding the number of communities in networks with arbitrary sparsity level.*

The last two decades saw a resurgence of interest in a problem popularly known as "community detection". A common problem definition is to partition $N$ nodes in a graph into $K$ communities such that there are differences in edge densities between within and between communities, where $K$ is assumed to be known *a priori*. Estimating number of communities ($K$) has recently become active in the literature. While the initial focus in the literature for estimating $K$ has been developing algorithms and drawing support from domain-specific intuition and empirical studies using the Stochastic Block Model (SBM), first proposed in Holland et al. (1983), (such as, Saade et al. (2014a), Yan et al. (2018)), there has been recent progress on attaining theoretical understanding of community numbers. Bickel & Sarkar (2015) and Lei et al. (2016) proposed hypothesis testing approaches based on principal eigenvalues or singular values. Some likelihood-based methods using the BIC criterion were proposed by Wang et al. (2017) and Hu et al. (2019). From a Bayesian perspective, Riolo et al. (2017) discussed priors for number of communities under the SBM and designed an Markov Chain Monte Carlo algorithm, Kemp et al. (2006) presented a nonparametric Bayesian approach for detecting concept systems, Xu et al. (2006) introduced an infinite-state latent

variable as part of a Dirichlet process mixture model, and Cerqueira & Leonardi (2020) proposed an estimator based on integrated likelihood for the SBM. Rosvall & Bergstrom (2007) introduced the concept of the minimum description length (MDL) to describe network modularities in partitioning networks, and Peixoto (2013) employed MDL to detect the number of communities. Chen & Lei (2018) and Li et al. (2020) proposed cross-validation based approaches with theoretical guarantees to estimate $K$. Yan et al. (2018) proposed a semi-definite programming approach, and Ma et al. (2018) proposed an estimator based on the loss of binary segmentation using pseudo-likelihood ratio. All of these approaches had theoretical guarantees. *However, all the theoretical results were obtained under the assumption that mean density of the networks is greater than* $\log(N)$.

Methods based on the spectrum of a certain class of matrices have become increasingly popular in recent years as non-parametric alternatives that are more computationally efficient and applicable to a wider range of settings. Most notably the non-backtracking matrices (e.g., Krzakala et al. (2013), Saade et al. (2014b), Coste & Zhu (2019), Bordenave et al. (2015), Saade et al. (2016)) and the Bethe Hessian matrices (e.g., Saade et al. (2015b), Lelarge (2018), Dall'Amico et al. (2019), Saade et al. (2015a), Dall'Amico et al. (2020), Saade et al. (2014a), Le & Levina (2015)) have received much attention due to their non-parametric form and competitive performance in the presence of degree heterogeneity and sparsity. In particular, unlike the non-backtracking operator, the Bethe Hessian is a real symmetric operator and hence offers additional computational advantages. Through simulations, Saade et al. (2014a) demonstrated that the Bethe Hessian outperformed the non-backtracking operator, belief propagation, and the adjacency matrices on clustering on both accuracy and efficiency. Le & Levina (2015) proved the consistency of the method based on the spectrum of the Bethe Hessian operator in semi-dense regimes, i.e., with the expected degree $\tilde{d} \gg \log(N)$ and the scalar parameter chosen from the two values commonly used in the literature based on heuristics for assortative and disassortative networks. *However, other than the two candidate values and their variations, there are no other known values for the scalar parameter to ensure the consistency result in any regime. Furthermore, real-world networks are generally much more sparse and there is no theoretical result in the literature that guarantees the effectiveness of the Bethe Hessian operator in more sparse regimes.*

**Our contribution.** In this paper, we contribute to the theoretical understanding of the Bethe Hessian operator in estimating $K$ for networks generated from the SBM in any regime regardless of the sparsity. We have three main contributions.

- We show that the method of estimating $K$ based on the spectral properties of the Bethe Hessian matrix ("spectral method") is statistically consistent, even in regimes more sparse than those previously considered in the literature, with the expected degree $1 \ll \tilde{d} \ll \log(N)$. The precise definition of $\tilde{d}$ is given in §2.1.

- We provide the first-of-its-kind interval of values for the scalar parameter of the Bethe Hessian operator that serves as a sufficient condition for the spectral method to correctly estimate $K$ asymptotically in network data.

- Through extensive simulations, we demonstrate that for any value chosen from the interval for the scalar parameter, the spectral method correctly estimates $K$ in networks regardless of sparsity. We also consider the heuristics-based values commonly used in the literature for the scalar parameter in the context of the interval.

The paper is arranged as follows. We present the definitions and a formal problem statement in §2. We present our main theoretical result and a sketch of the proof in §3, followed by empirical methods in §4. The simulation results and concluding remarks are given in §5 and §6, respectively.

## 2 PRELIMINARIES

### 2.1 NOTATION

An adjacency matrix, denoted by $\mathbf{A}$, is a random matrix whose rows and columns are labeled by nodes $i, j \in [N]$, where $\mathbf{A}_{ij} = 1$ if there is an edge between nodes $i$ and $j$ and 0 otherwise, and $[N]$ denotes the set $\{1, \ldots, N\}$. The mean observed degree is denoted by $\bar{d} := \frac{1}{N}\mathbf{1}_N^T \mathbf{A}\mathbf{1}_N$ and the

expected degree by $\tilde{d} := \frac{1}{N}\mathbf{1}_N^T \mathbb{E}\mathbf{A}\mathbf{1}_N$. $\lambda_\ell^\downarrow(\mathbf{A})$ denotes the $\ell$-th largest eigenvalue of $\mathbf{A}$ and $\lambda_\ell^\uparrow(\mathbf{A})$ denotes the $\ell$-th smallest eigenvalue of $\mathbf{A}$.

## 2.2 THE STOCHASTIC BLOCK MODEL

The stochastic block model (SBM) is a simple generative model for network data that embeds a community structure in an adjacency matrix $\mathbf{A}_{N \times N}$ of the randomly generated network. SBM has three parameters: (1) the number of communities $K$; (2) the membership vector $\boldsymbol{z} = (z_1, ..., z_N)$ that assigns a community label $z_i \in [K]$ to each node $i \in [N]$; and (3) the connectivity probability matrix $\mathbf{B}_{K \times K}$ where the element $B_{ab}$ represents the probability of an edge between nodes belonging to community $a$ and $b$, where $a, b \in [K]$.

$\mathbf{Z} \in \mathbb{Z}_{>0}^{N \times K}$ is defined as the community membership matrix such that $\mathbf{Z}_{ij} = 1$ if node $i$ belongs to community $j$ and 0 otherwise. We denote the maximum expected degree by $d_{\max} := N \max_i \sum_{j=1}^N [(\mathbf{ZBZ}^T)_{ij} - \text{Diag}(\mathbf{ZBZ}^T)_{ij}]$ and the maximum entry in matrix $\mathbf{B}$ by $d/N$, where $d := N \max_{a,b \in [K]} \mathbf{B}_{ab}$. $\lambda$ denotes the smallest eigenvalue of the normalized $\mathbf{B}$ matrix, $\lambda := \lambda_K^\downarrow\left(\frac{N}{d}\mathbf{B}\right)$. $\bar{\mathbf{A}}$ is the expectation of $\mathbf{A}$ and is computed as $\bar{\mathbf{A}} = \mathbf{ZBZ}^T - \text{Diag}(\mathbf{ZBZ}^T)$. $\bar{\mathbf{D}}$ is a diagonal matrix whose $i$-th diagonal entry is the sum of the $i$-th row of $\bar{\mathbf{A}}$. Let $\mathbf{N}$ be the vector of true community sizes and $N_{\min}$ denotes the number of nodes in the community with the lowest number of nodes in it.

A network generated from the SBM with parameters $K, \mathbf{B}, \mathbf{Z}$ is defined to be assortative if $B_{aa} > B_{ab}$ for all $a, b \in [K]$ with $a \neq b$, and if $\mathbf{B}$ has all positive eigenvalues (i.e., $\mathbf{B}$ has full-rank $K$). *The existing works in the literature on the spectral method referenced above have considered assortative networks, and we also consider assortative networks in this paper.*

## 2.3 THE BETHE HESSIAN MATRIX

The Bethe Hessian matrix associated with an adjacency matrix $\mathbf{A}$ is defined as

$$\mathbf{H}_\zeta := (\zeta^2 - 1)\mathbf{I}_N + \mathbf{D} - \zeta\mathbf{A} \tag{2.1}$$

where $\zeta > 1$ is a real scalar parameter, $\mathbf{D} := \text{Diag}(\mathbf{A}\mathbf{1}_N)$ is a diagonal matrix whose $i$-th diagonal entry corresponds to the degree of the $i$-th node, and $\mathbf{I}_N$ is an identity matrix of dimension $N \times N$.

As a real symmetric operator, $\mathbf{H}_\zeta$ is analytically tractable and computationally efficient, and has a number of useful properties. Saade et al. (2014a) demonstrated that the community structure in $\mathbf{A}$ can be recovered by applying a standard clustering algorithm (such as k-means clustering) to the eigenvectors of $\mathbf{H}_\zeta$ corresponding to negative eigenvalues. In the spectral clustering literature, those eigenvalues whose eigenvectors encode the community structure are known as the *informative eigenvalues* and have been observed to be well-separated from the bulk of the spectrum. In Saade et al. (2014a), $\zeta$ was set to be the square-root of the mean observed degree as a heuristic to render informative (negative) eigenvalues of $\mathbf{H}_\zeta$.

Le & Levina (2015) showed that the number of informative eigenvalues of $\mathbf{H}_\zeta$ directly estimate $K$ in the semi-dense regime ($\tilde{d} \gg \log(N)$) when $\zeta$ is set to be either $r_m := (d_1 + \cdots + d_N)^{-1}(d_1^2 + \cdots + d_N^2) - 1$ or $r_a := \sqrt{(d_1 + \cdots + d_N)/N}$. Both $r_m$ and $r_a$ are obtained based on heuristic arguments and are commonly used in the literature to estimate the radius of the bulk of the spectra. $r_a$ was considered in Saade et al. (2014a) and the choice of $r_m$ stems from the deep connection between the spectrum of $\mathbf{H}_\zeta$ and that of another matrix which is known as the non-backtracking operator $\mathcal{B}$. Denoting by $m$ the number of edges in $\mathbf{A}$, $\mathcal{B}$ is a $2m \times 2m$ matrix indexed by directed edges $i \to j$ and defined $\mathcal{B}_{i \to j, k \to l} = \delta_{jk}(1 - \delta_{il})$, where $\delta$ is the Kronecker delta and $m$ is the number of edges.

As in $\mathbf{H}_\zeta$, the informative eigenvalues of $\mathcal{B}$ are well-separated from the bulk of its spectrum and are real, so it also has been used to develop many popular non-parametric methods for clustering (see e.g., Saade et al. (2014b), Coste & Zhu (2019), Bordenave et al. (2015), Bruna & Li (2017), Gulikers et al. (2016)). This deep connection between $\mathbf{H}_\zeta$ and $\mathcal{B}$ was noted in Krzakala et al. (2013) and can be summarized by the phenomenon that, given any eigenvalue $\nu$ of $\mathcal{B}$, the determinant of $\mathbf{H}_\nu$ vanishes. However, unlike $\mathbf{H}_\zeta$, $\mathcal{B}$ is non-symmetric and its dimension ($2m \times 2m$) can get quite large. These present analytical and computational challenges when using $\mathcal{B}$, and in turn have popularized $\mathbf{H}_\zeta$ as a tool for clustering. Le & Levina (2015) showed that in semi-dense regimes

with expected degree $\tilde{d} \gg \log(N)$, the number of negative eigenvalues of $\mathbf{H}_\zeta$ directly estimate $K$ for $\zeta \in \{r_m, r_a\}$, where the methods were called BHm and BHa. In addition, it was noted that the number of negative eigenvalues of $\mathbf{H}_\zeta$ tend to underestimate $K$ when networks are unbalanced. Hence, corrections for $\mathrm{BH}_m$ and $\mathrm{BH}_a$ were proposed, namely BHmc and BHac, which heuristically estimate $\hat{K} = \max\{k : t\rho_{n-k+1} \leqslant \rho_{n-k}\}$ where $\rho_1 \geqslant \cdots \geqslant \rho_N$ are sorted eigenvalues and $t > 0$ is the hyperparameter. In light of this, we present the following problem we focus on in this paper:

**Problem Definition**: Suppose that we observe one network generated from the SBM, where the parameters $K, \mathbf{Z}, \mathbf{B}$ satisfy (i) assortativity, and (ii) the sparsity condition $\tilde{d} = o(\log(N))$. For the appropriate choices of $\zeta$, are the negative eigenvalues of the Bethe Hessian matrix $\mathbf{H}_\zeta$ still informative for estimating $K$? If so, what are the appropriate choices for $\zeta$? Can there be other heuristic choices for $\zeta$? Are the popular heuristic choices of $\zeta$, i.e., $r_m$ and $r_a$ as defined above (hereinafter "heuristic choices"), appropriate in the above sense?

## 3 THEORETICAL RESULTS

Our main contribution is twofold. First, we show that even in a sparse regime when $1 \ll \tilde{d} \ll \log(N)$, the number of informative eigenvalues of $\mathbf{H}_\zeta$ directly estimates $K$ consistently. Second, we provide the first-of-its-kind interval, which serves conveniently as a sufficient condition, of appropriate values for $\zeta$ for which the number of informative eigenvalues of the associated matrix $\mathbf{H}_\zeta$ directly estimates $K$. Below, we formally state this twofold result and provide a sketch of the proof, where we build intuition and provide key intermediate results. Precise statements and full proofs for all of the intermediate results discussed below are presented in §1.2 in the Supplement, along with statements and proofs of other relevant results in the literature.

**Theorem 3.1.** *(Main Result) Let $\beta := -d(\lambda N_{\min} - 1)/N$. For any $\delta \in (0, 3/2)$, $\mathbf{H}_\zeta$ has exactly $K$ negative eigenvalues for all $\zeta \in \frac{1}{2}\left(-\beta \pm \sqrt{\beta^2 + 4 - 4d_{\max}}\right)$ with probability at least $1 - \exp[-(\zeta/\sqrt{d})^{3/2-\delta}]$.*

*Sketch of the Proof* In assessing the spectral properties of $\mathbf{H}_\zeta$, it is more convenient to instead work with the spectrum of the associated Laplacian matrix, since it would allow us to use some of the important known results on the concentration of certain regularized adjacency matrix $\mathbf{A}$ around its expectation. Indeed, we are allowed to do so due to Sylvester's law of inertia (Theorem 1.4 in Supplement §1.1), which gives us that $\mathbf{H}_\zeta$ and the associated Laplacian have the same inertia. Note that the *inertia* of a real and symmetric matrix is a vector consisting of the number of positive, negative, and zero eigenvalues of the matrix.

To be more precise, consider the Laplacian $\mathbf{L}_\zeta := \frac{1}{\zeta}\mathbf{H}_\zeta = \tilde{\mathbf{D}}_\zeta - \mathbf{A}$, where $\tilde{\mathbf{D}}_\zeta = (\zeta - \frac{1}{\zeta})\mathbf{I}_N + \frac{1}{\zeta}\mathbf{D}$ and $\zeta > 1$. Now take its symmetric normalized version $\mathcal{L}(\mathbf{L}_\zeta) := \tilde{\mathbf{D}}_\zeta^{-1/2}\mathbf{L}_\zeta\tilde{\mathbf{D}}_\zeta^{-1/2}$. Then, by Sylvester's law of inertia, $\mathbf{H}_\zeta$ and $\mathcal{L}(\mathbf{L}_\zeta)$ have the same number of negative eigenvalues (see Lemma 1.5 in Supplement §1.2).

Next, to make the problem more tractable, we show that $\mathcal{L}(\mathbf{L}_\zeta)$ concentrates around its expectation $\mathcal{L}(\bar{\mathbf{L}}_\zeta)$ such that the problem can be stated in terms of the latter, which is a deterministic matrix, rather than the former, the random counterpart. More concretely, denote the expectation of the Laplacian $\mathcal{L}(\bar{\mathbf{L}}_\zeta) := \tilde{\bar{\mathbf{D}}}_\zeta^{-1/2}\bar{\mathbf{L}}_\zeta\tilde{\bar{\mathbf{D}}}_\zeta^{-1/2}$, where $\bar{\mathbf{L}}_\zeta = \tilde{\bar{\mathbf{D}}}_\zeta - \bar{\mathbf{A}}$ and $\tilde{\bar{\mathbf{D}}}_\zeta = (\zeta - \frac{1}{\zeta})\mathbf{I}_N + \frac{1}{\zeta}\bar{\mathbf{D}}$. Then, we decompose $\mathcal{L}(\bar{\mathbf{L}}_\zeta)$ into two parts. The first part is the difference between $\mathbf{A}$ and $\bar{\mathbf{A}}$, regularized by $\tilde{\mathbf{D}}_\zeta^{-1/2}$. The second part is the difference between $\bar{\mathbf{A}}$ regularized by $\tilde{\mathbf{D}}_\zeta^{-1/2}$ and $\bar{\mathbf{A}}$ regularized by the expectation of $\tilde{\mathbf{D}}_\zeta^{-1/2}$. In a regime satisfying $\tilde{d} = o(\log(N))$, the first part is bounded by $\frac{Cr^2}{\zeta - 1/\zeta}\left(\sqrt{d} + (\zeta^2 - 1)^{1/4}\right)$, where $C$ is a constant, with probability at least $1 - 2N^{-r}$ (see Theorem 1.3 in Supplement §1.1) due to a concentration result in Le et al. (2017) where it is shown that regularized $\mathbf{A}$ concentrates around its expectation. The second part is also bounded by $\frac{C'r^4\zeta^2}{\zeta^2-1}\left(\frac{d}{\zeta^2-1}\right)^3\left(1 + \frac{d}{\zeta^2-1}\right)^2$, where $C'$ is a constant, with probability at least $1 - e^{-2r}$ due to the properties of the Orlicz norm and Markov-Bernstein-type inequalities. Hence, for $\tilde{d} = o(\log N)$, the difference between the sample and its expected Laplacian, $\left\|\mathcal{L}(\mathbf{L}_\zeta) - \mathcal{L}(\bar{\mathbf{L}}_\zeta)\right\|$, is finite. Note

that this is a finite-sample result. We can obtain an asymptotic result from it by considering appropriate relationships among $d$, $\zeta$, and $r$, where $r \geqslant 1$ determines the probability $(1 - e^{-r})$ for the foregoing result. A sufficient condition for $\left\| \mathcal{L}(\mathbf{L}_\zeta) - \mathcal{L}(\bar{\mathbf{L}}_\zeta) \right\|$ to be $o(1)$ with high probability is $1 \ll r^{1/3} \ll \frac{\zeta}{\sqrt{d}}$ (see Lemma 1.6 in Supplement §1.2).

As the last step in this proof sketch, we apply Weyl's inequality to $\lambda_K(-\bar{\mathbf{L}}_\zeta)$ and $\lambda_{K+1}(-\bar{\mathbf{L}}_\zeta)$, and readily see that only the $K$ informative eigenvalues are negative, and hence the claimed result in the theorem (see Proof of Theorem 3.1 in Supplement §1.2). ∎

*Remark* (Theorem 3.1). Note that Theorem 3.1 is a finite sample result. The sufficient condition $\sqrt{d} \ll \zeta$ implies a high probability asymptotic result showing consistent estimation of the number of communities by the spectral method with $\zeta$ chosen from the interval given in the theorem. Hereinafter, we refer to the interval for $\zeta$ stated in Theorem 3.1 as the "oracle interval".

A sufficient threshold for detecting $K$ is presented below with a proof appearing in the Supplement.

**Corollary 3.2.** *In the setup of Theorem 3.1, with high probability, $K$ can be detected if the following threshold is satisfied:*

$$\lambda > \frac{2N\sqrt{d_{\max} - 1}}{dN_{\min}} + \frac{1}{N_{\min}} \tag{3.1}$$

## 4 EMPIRICAL METHODS

### 4.1 ESTIMATION OF THE INTERVAL FOR BETHE HESSIAN SCALAR PARAMETER

One practical consideration that needs to be addressed when implementing Theorem 3.1 and Corollary 3.2 is finding estimators of the parameters that are not directly observable in the data, namely $d$, $d_{\max}$, $\lambda$, and $N_{\min}$. Below, we propose an algorithm to compute the estimators for these oracle values. We do so by first estimating community memberships $\tilde{\mathbf{Z}}$ using regularized spectral clustering (Amini et al. (2013); Le et al. (2017)) and using maximum likelihood estimates to estimate the rest of the parameters. Then, the desired estimators are computed in a straightforward way.

---

**Procedure 4.1** PARAMS-ESTIMATION

**Input:** Adjacency matrix $\mathbf{A}$; a candidate number of communities $K_0$
**Output:** $\hat{\mathbf{N}}_{K_0}$: estimator for $\mathbf{N}$; $\hat{\mathbf{B}}_{K_0}$: estimator for $\mathbf{B}$; and $\hat{\mathbf{Z}}$: estimator for $\mathbf{Z}$

1: Obtain $\hat{\mathbf{Z}}$ using regularized spectral clustering of $\mathbf{A}$ with $K_0$ communities    ▷ See Remark (4.1)
2: $\hat{\mathbf{N}}_{K_0} \leftarrow \hat{\mathbf{Z}}^T \mathbf{1}_N$
3: $\hat{\mathbf{B}}_{K_0} \leftarrow \text{Diag}(\hat{\mathbf{N}}_{K_0})^{-1} \hat{\mathbf{Z}}^T \mathbf{A} \hat{\mathbf{Z}} \text{Diag}(\hat{\mathbf{N}}_{K_0})^{-1}$.

---

*Remark* (4.1). In Step 1, we need an algorithm which can consistently recover communities from $\mathbf{A}$. Other standard clustering algorithms can also be used in Step 1 as long as it consistently recovers community labels. The consistency of the estimators proposed in Algorithm 4.1 have already been established in Le et al. (2017). The time complexity of the procedure is $O(N^3)$ driven by the eigenvalue computation in Step 1. Hereinafter, we refer to the interval computed with the estimators from this procedure as the "estimated interval" (recall that the interval in Theorem 3.1 is referred to as the "oracle interval").

Procedure 4.1 outputs $\hat{\mathbf{N}}_{K_0}$ and $\hat{\mathbf{B}}_{K_0}$ with candidate number of communities $K_0 \in [1, ..., K_{\max}]$ as an input, where $K_{\max}$ is a tuning parameter. Then, the minimal community size is estimated with $\hat{N}_{\min} = \min\{\hat{\mathbf{N}}_2\}$. $\hat{N}_{\min}$ is an upper bound of $N_{\min}$ with high probability and has shown in simulations to be a good estimate of $N_{\min}$. Details on ad-hoc estimation of $d$, $d_{\max}$, and $\lambda$ using $\hat{\mathbf{N}}_{K_0}$ and $\hat{\mathbf{B}}_{K_0}$ and tuning parameter $K_0$ are given in the Supplement §1.3.

Figure 4.1 shows the simulation results on the performance of the oracle and estimated intervals for $\zeta$, and two popular heuristic choices $r_m$ and $r_a$. Under the setting of a large network ($N$) and the assortativity condition, the estimated intervals computed with Procedure 4.1 appear to match their oracle values well. It is shown that once the threshold in Corollary 3.2 is satisfied, $r_m$ and $r_a$

turn out to be sufficient, i.e., fall within the oracle interval. In §5, it is shown that values from the interval other than $r_m$ and $r_a$ can improve the performance, especially when $N$ is large in the sparse regime. Further extensive simulation results based on other parameter settings are included in the Supplement §1.3.

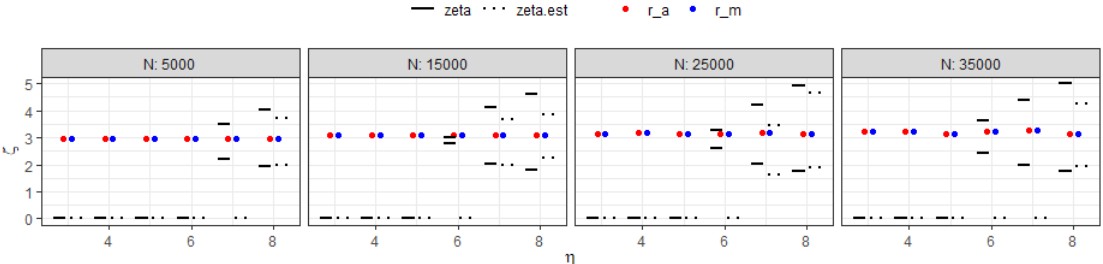

Figure 4.1: The oracle interval for $\zeta$ (Theorem 3.1) and its estimation (Procedure 4.1) are shown with two popular heuristic choices for $\zeta$ ($r_m$ and $r_a$) commonly used in literature. Network data was simulated from the SBM with the parameter settings shown in Table 5.1 with $K = 3$ and $\tilde{d} = 3\sqrt{\log(N)}$, each simulated with 20 repetitions. Intervals are shown as zeros when the threshold (3.1) is not met.

## 4.2 ESTIMATION OF THE NUMBER OF COMMUNITIES WITH THE BETHE HESSIAN

With a choice of $\zeta$ from the interval computed with the estimators from Procedure 4.1 above, we now propose an algorithm below that uses a spectral method to directly estimate $K$.

---
**Procedure 4.2** $K$-ESTIMATION
---
**Input:** Adjacency matrix $\mathbf{A}$; scalar parameter $\zeta$
**Output:** $\hat{K}$: estimator of $K$
  1: $\mathbf{D} \leftarrow \text{Diag}(\mathbf{A}\mathbf{1}_N)$
  2: $\mathbf{H}_\zeta \leftarrow (\zeta^2 - 1)\mathbf{I}_N + \mathbf{D} - \zeta\mathbf{A}$
  3: Obtain sorted eigenvalues $\lambda_1^\uparrow, ..., \lambda_N^\uparrow$ of $\mathbf{H}_\zeta$
  4: $\hat{K} \leftarrow \max\{k : \lambda_k^\uparrow < 0\}$
---

*Remark* (4.2). Just as with Procedure 4.1, the time complexity of Procedure 4.2 is $O(N^3)$ driven by the eigenvalue computation in Step 3.

Hereinafter, we refer to Procedures 4.1 and 4.2 as the "BHsparse" method.

## 5 EMPIRICAL STUDIES

We denote empirical accuracy rate (ACR) as the fraction of accurate estimates of $K$ out of 20 repetitions per simulation. Recent literature (Le & Levina (2015), Yan et al. (2018), Cerqueira & Leonardi (2020)) showed that methods based on the spectrum of the Bethe Hessian operator with popular heuristic choices for $\zeta$, i.e., $\{r_m, r_a\}$, are competitive in performance and computational efficiency in the semi-dense regimes. However, the synthetic networks used in the above references were relatively small (in terms of $N$) and more dense (with $\tilde{d} \gg O(\log(N))$) compared to the real-world networks. Through extensive simulations, we compare the performance of BHsparse with those based on the heuristic choices for $\zeta$ on large ($N$ up to 35,000) and sparse ($\tilde{d} = o(\log(N))$) networks. It is shown that BHsparse outperforms those based on the heuristic choices, especially as $N$ gets large and networks become more assortative.

## 5.1 DATA GENERATION AND SIMULATION SETTINGS

We simulate network data from the SBM under two different settings. In Simulation Setting (1), we define $\mathbf{B} := \rho \mathbf{B}_0 := \rho(\eta - 1)b[\mathbf{I}_K + \frac{1}{\eta-1}\mathbf{1}_K\mathbf{1}_K^T]$. $\rho$ controls the expected degree by $\tilde{d} = \rho(\mathbf{1}_N^T(\mathbf{Z}\mathbf{B}_0\mathbf{Z}^T - \text{Diag}(\mathbf{Z}\mathbf{B}_0\mathbf{Z}^T))\mathbf{1}_N)/N$. $\eta$ is the in/out ratio based on $\mathbf{B}$ and determines the degree of assortativity. $b$ is the baseline value in $\mathbf{B}$, which is set to $0.1$. We first simulate the membership vector $\mathbf{Z} \sim \text{Mult}\left(1; \left(\frac{1}{K}, ..., \frac{1}{K}\right)\right)$. We set $\tilde{d} \in \{3\sqrt{\log(N)}, 0.165(\log(N))^2, 0.788(N)^{(1/3)}\}$ by varying $\rho$, to assess the performance of the algorithms under different sparsity regimes. The constants in the rates of $\tilde{d}$ are chosen in way that $\tilde{d}$ is same at $N = 1000$ for all the rates. With a fixed $\mathbf{Z}$ and $\mathbf{B}$, and given model parameters $K$, $N$, $\tilde{d}$, and $\eta$, we then generate $\mathbf{A}$ with 20 repetitions. Table 5.1 summarises the combinations of model parameter settings used in the simulations.

Table 5.1: Model Parameters for Simulation Setting (1)

| $K$ | $N$ | $\tilde{d}$ | $\eta$ |
|---|---|---|---|
| 3 | $\{5000, 15000, 25000, 35000\}$ | $\{3\sqrt{\log(N)}, 0.165(\log(N))^2, 0.788(N)^{(1/3)}\}$ | $\{3, 4, ..., 8\}$ |
| 4 | $\{5000, 15000, 25000, 35000\}$ | $\{3\sqrt{\log(N)}, 0.165(\log(N))^2, 0.788(N)^{(1/3)}\}$ | $\{3, 4, ..., 8\}$ |
| 10 | $\{5000, 15000, 25000, 35000\}$ | $\{3\sqrt{\log(N)}, 0.165(\log(N))^2, 0.788(N)^{(1/3)}\}$ | $\{16, 17, ..., 25\}$ |
| 25 | $\{25000\}$ | $\{3\sqrt{\log(N)}, 0.165(\log(N))^2, 0.788(N)^{(1/3)}\}$ | $\{41, 42, ..., 55\}$ |
| 50 | $\{25000\}$ | $\{3\sqrt{\log(N)}, 0.165(\log(N))^2, 0.788(N)^{(1/3)}\}$ | $\{101, 102, ..., 110\}$ |

In Simulation Setting (2), we use a more general probability connectivity matrix as defined in equation 5.1, where $\eta \in \{2.5 + (m - 1)0.25 : m = 1, ..., 9\}$, and set other parameters as follows: $\tilde{d} = 3\sqrt{\log(N)}$; $K = 3$; and $N \in \{5000, 15000, 25000, 35000\}$.

$$\mathbf{B} := \rho \begin{pmatrix} 1+\eta & 0.5 & 0.3 \\ 0.5 & 2+\eta & 0.1 \\ 0.3 & 0.1 & 0.5+\eta \end{pmatrix} \tag{5.1}$$

## 5.2 SIMULATION RESULTS

Figure 5.1 below shows ACR of BHsparse versus $\eta$, with varying values for $\zeta$ chosen from quantiles of the oracle interval in Theorem 3.1. It is clear that there is a threshold value of $\eta$ below which detection of $K$ fails and otherwise it succeeds. The top row (A) shows that this threshold decreases as $N$ increases from $5,000$ to $15,000$ while the bottom row (B) shows that the threshold increases with $K$. Note that the threshold for $\lambda$ in equation 3.1, which depends on $\eta$, decreases as $N$ increases.

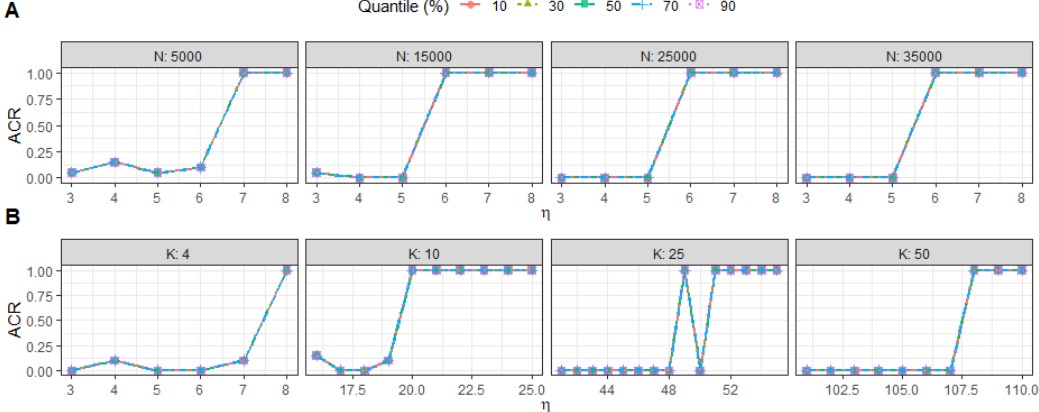

Figure 5.1: ACR of BHsparse with $\zeta$ set to quantiles (10%, 30%, 50%, 70%, 90%) of the oracle interval in Theorem 3.1. Network data was generated from Simulation Setting (1) with fixed $\tilde{d} = 3\sqrt{\log(N)}$. (A) shows ACR versus $\eta$ for varying levels of $N$ with $K = 3$. (B) shows ACR versus $\eta$ for varying levels of $K$ with $N = 25,000$.

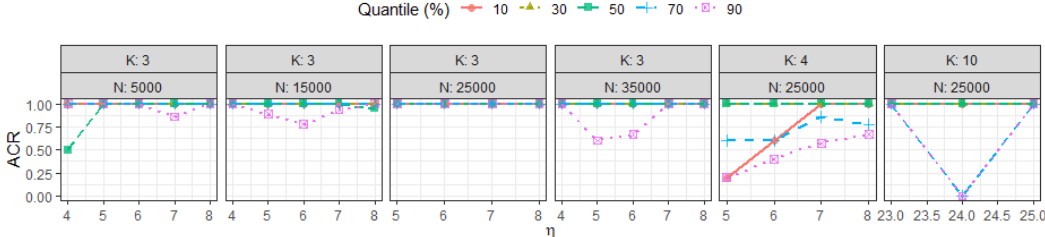

Figure 5.2: ACR of BHsparse versus $\eta$ as $K$ and $N$ vary, using estimated intervals with $\zeta$ set to quantiles (10%, 30%, 50%, 70%, 90%) of the estimated intervals using Procedure 4.1 based on networks satisfying the threshold (3.2). Network data was generated from Simulation Setting (1) with fixed $\tilde{d} = 3\sqrt{\log(N)}$.

Figure 5.2 shows ACR of BHsparse with $\zeta$ set to different quantiles of the estimated intervals. Only those cases where either interval exists are shown in the plot. It can be observed that the performance becomes worse as $\zeta$ gets close to end-points of the intervals. Generally 30% to 50% quantiles within the intervals appear to work the best.

In Figure 5.3 (Figure 5.4 resp.), we compare the performance of BHsparse using 30% and 50% quantiles of oracle intervals (estiamted intervals resp.) with BHmc and BHac. Figure 5.3 and 5.4 show that when the threshold in Corollary 3.2 is satisfied, $\zeta \in \{30\%, 50\%\}$ quantiles of both the oracle and estimated intervals perform better than the two heuristic choices in Le & Levina (2015).

The plots corresponding to Figures 5.1, 5.2, 5.3, and 5.4 for the other two density regimes of $\tilde{d} \in \{0.165(\log(N))^2, 0.788(N)^{(1/3)}\}$ are given in the Supplement §1.3.

We also compare performances of BHsparse of $\zeta$ equals 30%, 50%, and 70% quantiles of the estimated intervals with BHmc and BHac with a more general setting of the probability connectivity matrix as Equation 5.1. Figure 5.5 shows the ACR performances of our proposed method with choices of $\zeta$ as 30% to 50% quantiles of the intervals over-perform the methods proposed in Le & Levina (2015).

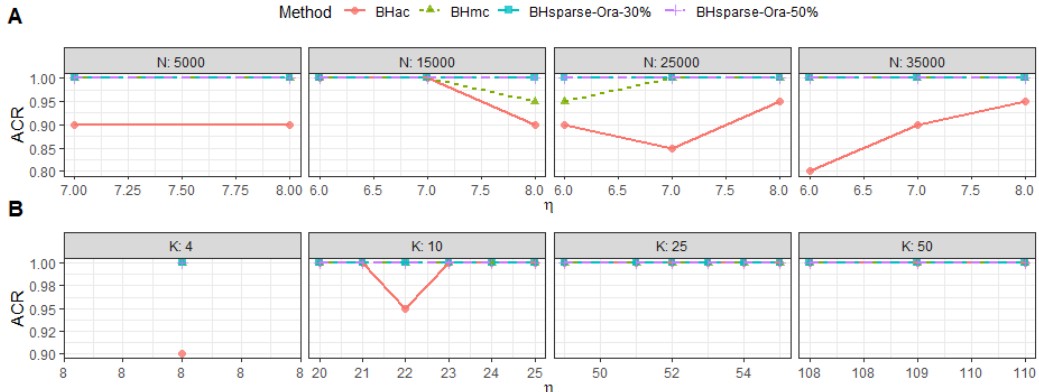

Figure 5.3: Row (A) shows ACR versus $\eta$ using oracle intervals, with different values of $N$ and $K = 3$. Row (B) shows ACR versus $\eta$ as $K$ varies with fixed $N = 25,000$. Both plots only include cases where oracle thresholds in Corollary 3.2 are satisfied and are based on data generated from Simulation Setting (1) with fixed $\tilde{d} = 3\sqrt{\log(N)}$.

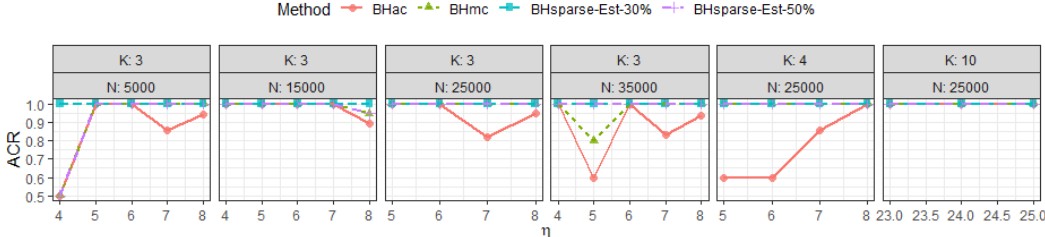

Figure 5.4: ACR versus $\eta$ as $K$ varies using estimated intervals, based on data from Simulation Setting (1) with fixed $\tilde{d} = 3\sqrt{\log(N)}$ and only including cases where estimated thresholds in Corollary 3.2 are satisfied. For $\zeta$, 30% and 50% quantiles of the estimated intervals are considered.

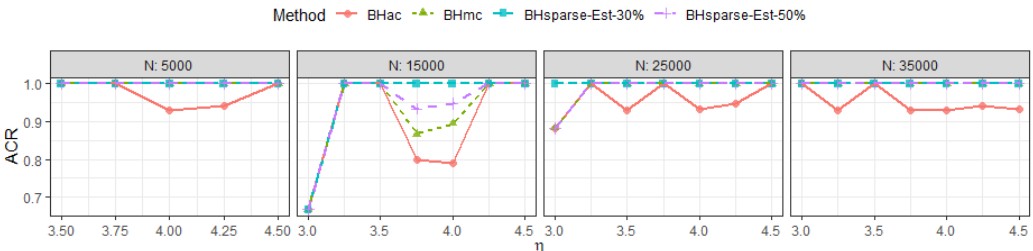

Figure 5.5: ACR versus $\eta$ as $N$ varies using estimated intervals, based on data from Simulation Setting (2) with $K = 3$ and $\tilde{d} = 3\sqrt{\log(N)}$, and only include cases where estimated thresholds in Corollary 3.2 are satisfied. For $\zeta$, 30% and 50% quantiles of the estimated intervals are considered.

### 5.3 Real-World Network Application

We apply our proposed methods to a benchmark real-world network data set, the Polbooks network, which also been used in previous works (Le & Levina (2015), Chen & Lei (2018)). The Polbooks network (Rossi & Ahmed (2015)) represents books on politics published around the 2004 presidential election that were purchased together as a bundle from Amazon.com. The Polbooks network has two natural communities (liberal and conservative), 105 nodes, and the mean observed degree of 8.4. The estimated interval for $\zeta$ using Procedure 4.1 is [1.3, 7.3]. Using the 50th percentile of the interval, Procedure 4.1 correctly estimates the number of communities, $K$, as $\hat{K} = 2$, while both BHac and BHmc estimate the number of communities, $K$, as $\hat{K} = 4$. This result demonstrates that Procedures 4.1 and 4.2 can correctly detect the number of communities in real-world networks too.

## 6 Discussion

In this paper, we contribute theoretical results on the selection of Bethe Hessian scalar parameter, $\zeta$, for a consistent estimation of number of communities ($K$) in networks that are generated from the SBM with arbitrary degree of sparsity. To the best of our knowledge, this is the first study to theoretically prove the consistency of the Bethe Hessian spectral method to estimate $K$ in sparse regimes with $\tilde{d} = o(\log(N))$. We also rigorously derive the oracle interval and provide a convenient way to empirically estimate the intervals for selecting $\zeta$ to construct the Bethe Hessian operator to consistently estimate $K$. We support our theoretical results with simulation studies and real-world network application too.

In this paper, we only prove an upper bound of the hypothesized threshold for estimation of number of communities. An important future work will be to prove the lower bound results such that the existence of the threshold can be properly established.

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
