# OpenReview forum: "Estimation of Number of Communities in Assortative Sparse Networks"
_ICLR.cc/2021/Conference — Reject_

### Official Review · AnonReviewer1 · 2020-10-27
**Interesting approach with too limited experimental validation to assess usefulness**

**Rating:** 6
**Confidence:** 4

**Review:**

Summary
The authors propose a spectral framework using the Bethe Hessian matrix to infer the number of communities in sparse networks. The method relies on the eigendecomposition of the Bethe Hessian matrix for which negative eigenvalues are preserved and the number of such eigenvalues used to define the number of communities. In particular, theoretical guarantees for settings of the scalar of the Bethe Hessian matrix is derived including an associated spectral estimation procedure.

Pros: The procedure is very easy to implement and extends existing work to also operate in the sparse regime of d=o(log(N)) which is much more relevant and realistic from a practical perspective.

The manuscript includes nice theoretical contributions including proofs that the normalized Laplacian form of the Bethe Hessian preserves inertia, that the normalized Laplacian concentrates around its expectation, and bounds on the recovery of number of communities for suitable intervals of the scalar paramter of the Bethe Hessian matrix.

Cons: The k-means procedure is not guaranteed to identify the same clustering configuration each time and thus the robustness of the procedure hinges on the robustness of k-means (which is not particularly robust). That being said this is not specifically just an issue of this procedure but in general when using spectral approaches relying on a subsequent clustering step. It would be good to furher discuss robustness in regards to procedure 4.2.

There is a large body of literature on identifying the number of clusters for the SBM and several key references within Bayesian inference in this context is missing. This includes the non-parametric SBM called the infinite relation model considering sampling for inference
Kemp, Charles, et al. "Learning systems of concepts with an infinite relational model." AAAI. Vol. 3. 2006.
Xu, Zhao, et al. "Learning infinite hidden relational models." Uncertainty in Artificial Intelligence (UAI2006) (2006): 2.
and variational inference.
Xu, Zhao, et al. "Fast Inference in Infinite Hidden Relational Models." MLG. 2007.
Further work includes the use of minimum description length to quantify complexity:
Rosvall, Martin, and Carl T. Bergstrom. "An information-theoretic framework for resolving community structure in complex networks." Proceedings of the National Academy of Sciences 104.18 (2007): 7327-7331.
The impact of the assumption that B in the stochastic block model is full rank and assortative is unclear. Typically in SBM there are no such restrictions and it would be good to argue for the validity of this assumption. Clearly, a very community structured network will be diagonal dominant and in general we can expect the rank to be full, but it would be good to clarify the impact of this assumption. The synthetic study is very simple including a within and between link density structure (two-parameters in the B matrix) – it would be good to elaborate on how the approach would work considering more complex structured B matrices.
The paper only includes synthetic analyses and it is unclear how practical useful the procedure is. It would strengthen the paper to also demonstrate the approach on real data and how community structures are inferred and differ from existing procedures in the real network regime.

Quality
The paper is in general well written and clear. The mathematical derivation is non-trivial relying in particular on work by (Can M Le et al., 2017)  but the experimentation are somewhat lacking. The approach is solely contrasted existing procedures using the Bethe Hessian and not the large existing literature proposing various means to estimate the number of components within the SBM formulation including the literature also reviewed in the introduction. Furthermore, only a very simple synthetic dataset study is executed in which the SBM is restricted to a within community density and between community density for which the contrast $\eta$ between within and between density is varied. It is therefore unclear what the merits of the present procedure is in contrast to existing methods for inferring the number of communities. It would strengthen the paper to consider also real networks for instance as in (Le & Levina, 2015) considering networks with “Ground Truth” (see also https://arxiv.org/pdf/1507.00827.pdf).

Clarity
To make the paper more self-contained it would be good to explain the structure of the Bethe matrix and how it is motivated as well as the role of the scalar parameter >1.
Minor typos:
excellent excellent –> excellent
I suspect line 3 in Procedure 4.2 should have Z 1_N -> Z^T 1_N.

Originality
The approach is based on results for the semi-sparse regime (i.e. average degree >> log(N)) but presently developed for the sparse regime, i.e. average degree = o(log(n)) with theoretical results for an interval of the scalar parameter used when specifying the Bethe Hessian matrix as well as a simple (spectral) algorithm to infer the communities and their number and a spectral procedure for defining an initial B0 matrix and community size vector N0 used to define plausible intervals for the scalar parameter of the Bethe Hessian matrix.

Significance
The paper extends the Bethe Hessian approach to the sparse domain and provides simple algorithmic procedures for inferring communities based on the spectral decomposition of the Bethe Hessian and a heuristic procedure setting the scalar for the Bethe Hessian according to an interval inferred through a procedure defining plausible estimates of the between community connectivity matrix B. The approach is only evaluated on very simple synthetic problems with very clear simple community structure not contrasting the procedure to any alternatives but approaches relying on the Bethe Hessian. As such, the significance and impact of the produced method is at this point very unclear and hard to assess. It is also unclear to what extend the results generalize beyond the synthetic experiments including an c_in and c_out link density to more general stochastic block modeling in which the B matrix has separate elements for each entry.

In summary, I consider the manuscript right around the acceptance threshold. The derivations are solid and sound, but the paper does not demonstrate the practical utility on real networks considering only very simple synthetic networks. This leaves me wondering if the approach would actually work in a practical setting and in particular, if the procedure for estimating the scalar interval based on procedure 4.2 can be used in practical applications.


______________________________________ Updates in regards to the authors' response__________________________________________________________

I appreciate the authors efforts to improve the manuscript and I think the manuscript has improved. I therefore raise my score to marginally above acceptance.

In the abstract computational efficiency is highlighted but from the response it seems the approach scales as O(N^3) this is not particularly efficient as many procedures such as conventional SBM can exploit network sparsity for computational scaling. I think stating the approach as efficient is somewhat misleading based on the response explicitly made to Reviewer #4.

I appreciate the added synthetic analyses and that a real network analysis is included but I am somewhat disapointed that only one real network is considered as opposed to including a series of existing networks with ground truth community structure. I thus find that the experimental validation on actual networks could be further strengthened - but it is good to see a real result.


Minor comments:

When stating "All of these approaches had theoretical guarantees." I believe this refers to some of the approaches discussed and not all. It would be good to clarify which approaches.

estiamted -> estimated

which also been used -> which has also been used

---

### Official Review · AnonReviewer3 · 2020-10-29
**THe authors show an algorithm using the spectrum of the Bethe-Hessian matrix to determine the number of communities of a graph generated according to SBM.**

**Rating:** 6
**Confidence:** 3

**Review:**

I really like this paper. But there are a few questions that I have which are not clear to me.

1)  What is the threshold for detecting if the number of communities K is 2 or more than 2?
2) What does equation 3.3 mean? If the threshold is satisfied, then we can infer the number of communities? Should the threshold not depend on K?
3) It is also unclear from previous literature if the work presented by the authors is incremental or not. I would like to see a paragraph explaining the main difficulties while proving these results given the previous work.

However, this work has a number of pros

1) The paper provides a very nice theoretical guarantee on when it will be possible to detect the number of clusters
2) The algorithm and the theoretical lemmas are stated rigorously.

Cons

1) I would have really liked to see a sketch of the proofs of the lemmas in the main draft. It would provide intuition on why the spectrum of the Bethe-Hessian matrix is used to determine the number of components.

2) The authors have not discussed any lower bounds beyond which the number of communities cannot be detected

3) Is it possible to detect the number of communities approximately? Surely, it would be possible to detect a single large community. Maybe the authors can discuss approximate detection of communities where failure to detect a community has a cost proportional to its size.

But I believe that the pros of this paper outweigh its cons and therefore I recommend acceptance.

---

### Official Review · AnonReviewer2 · 2020-10-29
**Interesting paper with some work needed.**

**Rating:** 7
**Confidence:** 4

**Review:**

Post Rebuttal:

I think the paper is much improved, and I'm bumping up my score accordingly.

Small point - in corollary 3.2, to say 'with high probability' still requires that explicit conditions such as $|\beta|/\sqrt{d} \to \infty$ are stated. Thm 3.1 makes no claim of _high_ probability - it instead has an explicit bound, and does not impose conditions needed to make this bound 'high'. As a conseuqnce, the conditions of Thm 3.1 do not suffice for corollary 3.2.

----


Summary:

The paper studies the estimation of the number of communities in a graph drawn from a stochastic block model (SBM) using spectral properties of the Bethe Hessian of the observed graph. While this scheme has been proposed in the prior literature, the focus here is on sparse graphs, with average degrees bounded as $o(\log N)$. The authors show a sufficient condition for consistency of this procedure in this regime, and provide simulations as evidence for the effectiveness of the resulting method. These simulations incorporate the challenges of having to estimate the various parameters of the generative model.

---

Strengths:

The problem is contexaulised well, and related work is adequately discussed and compared to. The theoretical analyses provided are crisp, and use existing results on concentration of random graphs in a nice way. I particularly appreciate that the simulations dealt with the fact that parameters of the models need to be estimated. The procedure to do so is well described, with the performance gains clearly demonstrated for small $K$.

Weaknesses:

For me there are two main weaknesses:

a) Section 3 needs some polish.

First, The section is begun by directly launching into a proof sketch for the relevant method. This comes out of nowhere, and would be helped by describing the precise main result before launching into a proof of it. This could be done, perhaps, by stating Thm 3.3 before the argument, and then launching into the proof sketch.

Second, I don't think corrolary 3.4 is entirely accurate. There's a minor issue of an additive $(d/N)$ term that's missed on the right hand side. More importantly, in order to show consistency in the usual setting of large graphs that is taken in SBMs, you need to be able to drive the probability of error to $0$, which requires that $\zeta/\sqrt{d} \to \infty$ as $N$ blows up. The necessary condition stated doesn't allow that _unless_ $d_{\max}$ blows up with $N$. This is worth saying since the result of this corollary is ultimately compared to weak recovery of the $2$-SBM, where the claim made is not valid because the condition for weak recovery that is stated holds even when $d_{\max}$ and $d$ are $O(1)$.

Speaking of, I want to point out that the condition on the top of page 4 is for _weak recovery_, that is, the problem of obtaining a partition which is _slightly_ better than random guessing. The phrasing of the paragraph suggests that the problem being compared to is that of exact recovery, for which the SNR threshold is very different (e.g. degrees need to blow up logarithmically). Also, it is worth noting here that even the distinguishability problem for the SBM (i.e., hypothesis testing between an SBM and an unstructured Erdos-Renyi graph of the same average degree) in the $2$-SBM requires $(a-b)^2 > 2(a+b)$. This is a weaker problem than recovering the number of communities, and so effectively serves as a lower bound. See, e.g. [1].

Minor aside - it might be better to state Corollary 3.4 in terms of $\lambda$ instead of $\lambda_K^{\downarrow}$. The reason for this is that as it stands, it looks like the left hand side grows linearly with $N$, while the right hand side grows sublogarithmically. This is, of course, not the case because $\lambda_K^{\downarrow}$ itself must behave at the scale $o(\log N)/N$ in order for the degrees to be $o(\log N)$. The normalised $\lambda$ gets rid of this effect.

b) In my opinion, the experiments need to be deepened.

Ultimately the result given can be seen as a thresholding of $\frac{d\lambda}{\sqrt{d_{\max} -1}} \frac{N_{\min}}{N},$ or, in the multinomial prior chosen, in terms of $\frac{d\lambda}{\sqrt{d_\max}-1} \frac{1}{K}$ (with high probability, up to small corrections). The experiments vary the first term by altering $\eta$, but the values of $K$ explored are very small - only $K = 3$ and $4$ are explored. This effectively means that the behaviour of the scheme with respect to the ground truth $K$ is left largely unexplored. (It is claimed that Fig 5.1 shows the behaviour with respect to $N_{\min}$ but this is not the case - for instance, $N_{\min}$ increases by a factor of $1.75$ when going for Fig 5.1 b) to c), but the threshold remains the same. I think what's actually being seen is the limiting effect in that the data aligns more strongly with the large graph limit as you increase $N$.)

In fact, the behaviour with respect to $K$ has twofold importance. First, it enters as a parameter of the threshold that is developed, and this needs to be empirically confirmed. Secondly, a critical issue in many community SBMs is the behaviour of the achievability thresholds as $K$ grows large relative to $N$ - for instance, for large $K$, the Kesten-Stigum theshold and the information theoretic thresholds separate, and a computational-statistical gap is conjectured for weak recovery. Thus exploring how the scheme empirically behaves as $K$ grows is of technical interest.

Minor comment - in section 5.1, it might be worth explicitly mentioning that you set $\tilde{d} = 3\sqrt{\log N}$ and vary $\rho$ to make the average degree come out correct. Also, it would be valuable to explicitly show the threshold condition of Corollary 3.4 in terms of these new parameters.

---

Overall impressions:

I quite like this paper. The problem is certainly relevant, and the analyses provided are the first, in my knowledge, to explore the sublogarithmic degree regime in the recovery of the number of communities in a graph with planted partitions with both theoretical and parameter-aware simulation studies. The method is interesting and well fleshed out.  I think the paper does have some flaws - most critically the empirical study with respect to $K$, and the issues I mentioned regarding Corollary 3.4 (which is a bit oversold as of now). These are the main bottlenecks in my current rating of marginally below acceptance. I can fully see myself bumping the rating to 7 if they are dealt with.

---

Nitpicking:

- Operators in mathmode should be escaped via ``````\operatorname {XYZ}, which gives $\operatorname{Diag}(A)$ instead of $Diag(A)$. Similarly \min, \max, \log are available as mathmode commands, giving $N_{\min}$ and not $N_{min}$.

- I'm not sure that the line plots in section 5 are the best way to present this data, mainly because there are so few data points. I'm also not sure what should be used instead, so this is not totally constructive, but it may be worth exploring other ways (maybe just a table?). Also, i have no clue what CND.MTD.correct means.

[1]: Banks, J., Moore, C., Neeman, J., & Netrapalli, P. (2016, June). Information-theoretic thresholds for community detection in sparse networks. In Conference on Learning Theory (pp. 383-416).

---

### Official Review · AnonReviewer4 · 2020-10-30
**Well-motivated problems and relevant to the ML community. Non-trivial way to configure the methods. Experiments need improvement.**

**Rating:** 5
**Confidence:** 2

**Review:**

In this paper the authors consider the problem of computing the number of communities K in an arbitrarily sparse graph generated under the Stochastic Block Model (SMB). Previous studies that consider the problem of computing K show theoretical guarantees only in graphs with average degree $\Omega(\log n)$. One of the previous studies (namely, [1]) has shown that  the number of communities equals the number of negative eigenvalues of the Bethe Hessian matrix for graphs with expected average degree $\Omega(\log n)$ under SBM. In this paper the authors show that with an appropriate scalar parameter for the Bethe Hessian matrix of graphs with average degree $o(\log n)$, the same property still holds, and thus obtain a method for computing K in graphs of sublogarithmic density. In particular, the authors give an interval for the choice of the $\zeta$ scalar that depends on several parameters of the underlying SBM distribution.

To use this result in practice (i.e., to set the right $\zeta$), one has to estimate the maximum expected degree of the underlying SBM distribution, the maximum probability of an edge, and the smallest eigenvalue of the normalized probability matrix B of the SBM. However, this is a non-trivial task, and the authors show that using estimates (with no guarantees) of B and the sizes of the communities they get reasonable estimates in practice and can thus choose a reasonable parameter $\zeta$.

Finaly, the authors compare their choice of $\zeta$ with the one from [1] and show that on graphs with sparsity $3\sqrt{\log n}$ and sufficient assortativity the Bethe-Hessian-based method more frequently estimates the right number of communities.

=================
PROS
+ The problem of computing the number of communities is important and relevant to the ML community, as most methods for computing the actual communities require prior knowledge of K.
+ This is the first study to show theoretical guarantees in the sublogarithmic density regime in graphs generated by the SBM. As the authors stretch out this is an important practical scenario.

CONS
- The time complexity of the method and the estimation computation is not discussed.
- Non-extensive experiments (K is set only to 3 and 4, and the average degree is set to a single value)
- The right choice of the parameter $\zeta$ is non-straightforward, and the authors suggest non-guaranteed estimations. That makes the theoretical guarantee of the overall method weaker, as one has no guaranteed way of tuning it.
The problem itself is well-motivated and the regime of sublogarithmic average degree is important, and typically real-world graphs fall under this category.

The presentation of the paper is acceptable, although I would like to see some further discussions:* Please add a formal definition of the problem and explain the connection of the number of the negative eigenvalues of the Bethe Hessian matrix and the number of communities. Unless I missed something, this is not discussed.* I think it would be beneficial to discuss the time complexity of the method. That would also support the claim about efficiency.

The choice of $\zeta$ is non-straightforward. Recall that $\zeta$ needs to be set right to enable the theoretical guarantees of the method. My understanding is that the authors rely on heuristics to estimate the right parameters and eventually choose the appropriate $\zeta$. If there is no guaranteed way of computing the right $\zeta$, then in practice one has no way to run the algorithm with guarantees. Please, do correct me if I'm wrong here.

The experimental section could also be improved. My main complaint is the very limited number of communities generated under the SBM. In particular, the authors only consider setting $K \in {3,4}$, and this is not justified. Why not set K = 2, 5, 10, 50, 100 and show the performance (also compared to [1]) of the method as K changes? Also, it would be nice to consider different densities ranging from constant average degree to super-logarithmic. My understanding is that there is no upper bound on the densities supported by the method.

Overall, I'm not enthusiastic about the paper because of the non-systematic way of choosing $\zeta$, and because of the weak experimental evaluation.

Minor things:
* there are some hats flying around in Procedure 4.2, Line 4
* " expected expected degree" in page 6

[1] Can M Le and Elizaveta Levina. Estimating the number of communities in networks by spectral methods. arXiv preprint arXiv:1507.00827, 2015.

---

### Decision · Program_Chairs · 2021-01-07
**Final Decision**

**Decision:**

Reject

**Comment:**

This paper describes a procedure for estimating the number of clusters in assortative sparse networks generated from a stochastic blockmodel, where the average degree scales sublogarithmically with the number of nodes. The approach proposed by the authors is based on the spectra of the Bethe Hessian matrix.

The article is well written. The reviewers raised a number of questions regarding the ad-hoc procedure for estimating the parameter $\zeta$ needed to estimate $K$, and the limited experiments. While the authors provided some additional experiments in the revised version, including on a real world dataset, the overall article stills appears to be too borderline for ICLR. Adding experiments on other real-world datasets would strengthen the paper.

I recommend rejection.